# TubeDAgger: Reducing the Number of Expert Interventions with Stochastic Reach-Tubes

## Abstract

Interactive Imitation Learning deals with training a novice policy from expert demonstrations in an online fashion. The established DAgger algorithm trains a robust novice policy by alternating between interacting with the environment and retraining of the network. Many variants thereof exist, that differ in the method of discerning whether to allow the novice to act or return control to the expert. We propose the use of stochastic reachtubes - common in verification of dynamical systems - as a novel method for estimating the necessity of expert intervention. Our approach does not require fine-tuning of decision thresholds per environment and effectively reduces the number of expert interventions, especially when compared with related approaches that make use of a doubt classification model.

## 1 Introduction

Imitation learning (IL) offers a practical framework for training autonomous agents by mimicking expert behavior. While supervised methods such as behavioral cloning Pomerleau (1988) are simple to implement, they suffer from compounding errors due to covariate shift: the trained policy may visit states at test time that deviate from the expert's distribution, where it is likely to perform poorly. To mitigate this, *interactive imitation learning* algorithms have been proposed, most notably DAgger Ross et al. (2011), which reduces covariate shift by iteratively collecting data from the policy's own rollouts while querying the expert for corrective actions.

Despite their effectiveness, DAgger-style approaches raise practical concerns in real-world systems, where expert interventions can be expensive, time-consuming, or safety-critical. SafeDAgger Zhang & Cho (2017) addresses this by introducing a safety (or "doubt") model to predict when the policy is likely to deviate from the expert, allowing the system to fall back to the expert only when necessary. LazyDAgger Hoque et al. (2021) further reduces unnecessary switching by adding a hysteresis mechanism and injecting noise into expert actions to encourage policy robustness. However, both approaches rely on a learned classification model, which may introduce additional difficulties during training.

In this paper, we introduce **TubeDAgger**, a novel interactive imitation learning algorithm that uses *stochastic reachability analysis* for creating a decision boundary that is independent of the learner's experience. TubeDAgger constructs a stochastic reach-tube before the start of training, and delegates control to the expert only when the experienced states exceed a specified safety threshold in relation to the reachable set. This leads to fewer expert interventions while maintaining strong policy performance. Intuitively speaking, a learning model can remain in control - even if it is acting differently than the expert - as long as it is experiencing a familiar trajectory of observations.

The contributions of our work are as follows:

- We introduce TubeDAgger, a new interactive imitation learning algorithm that uses stochastic reachtubes to reduce the number of expert interventions.

- We demonstrate that TubeDAgger eliminates the need for training and maintaining a separate doubt classification model, simplifying training while maintaining safety guarantees.

- We empirically evaluate TubeDAgger across multiple locomotion tasks, demonstrating significant reductions in expert intervention frequency while maintaining task performance.

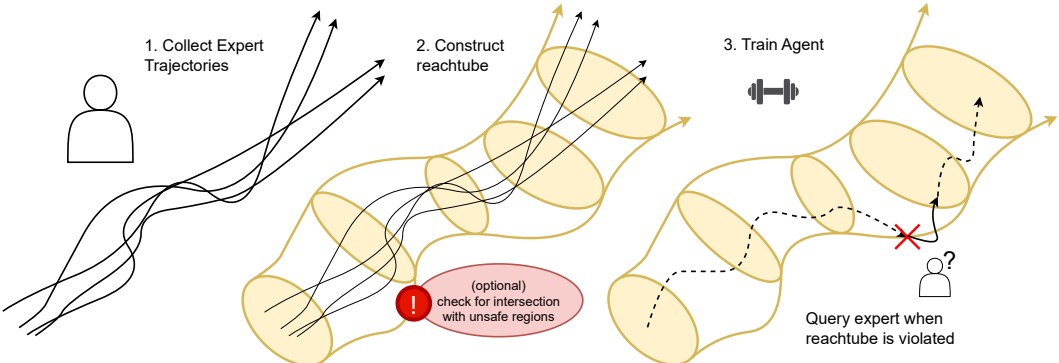

Figure 1: A schematic of the TubeDAgger approach which encompasses initial collection of expert trajectories, construction of a reachtube, and employing it as a decision boundary in place of the doubt model of LazyDAgger.

Figure 1 shows a schematic of our proposed approach. First, expert demonstrations are collected. They are then used to construct a stochastic reachtube. Optionally, a check for intersection with known unsafe states can be performed, prompting the collection of more expert data if unsuccessful. The reachtube is then used as a decision boundary for expert interventions.

The remainder of this paper is organized as follows: Section 2 provides necessary background on reachability analysis and interactive imitation learning. Section 3 details our TubeDAgger algorithm and its implementation. Section 4 describes our experimental setup and results. Section 5 discusses related work in imitation learning and safety-critical control. Finally, Section 6 concludes with discussions on limitations and future work.

## 2 BACKGROUND

### 2.1 INTERACTIVE IMITATION LEARNING

**SafeDAgger.** In the original DAgger framework Ross & Bagnell (2010), the training process gradually shifts control from the expert to the novice policy. Initially, actions are exclusively taken by the expert, and over time, the probability of using the novice policy increases linearly. However, a significant drawback of this approach is the potential for the novice to suggest unsafe or catastrophic actions – especially in the early stages of training when it is poorly trained. SafeDAgger Zhang & Cho (2017) addresses this safety concern by constraining the novice to act only when it is sufficiently close to the expert. Specifically, it introduces a mechanism to assess whether the novice's predicted action $\hat{\pi}(s)$ – at state $s$ – is within a certain distance $\tau_m$ of the expert's action $\pi^*(s)$. Formally, equation 1 should hold at all times for the novice behavior to be considered *safe*. For measuring the distance, typically the $\ell_2$ norm is used denoted by $||.||_2$.

$$||\hat{\pi}(s) - \pi^*(s)||_2 < \tau_m \tag{1}$$

In practice, a separate safety (or "*doubt*") model is trained to predict whether the novice policy, given a particular state, will produce an action that lies within the predefined threshold $\tau_m$ of the expert's action. At runtime, this doubt model serves as a gatekeeper: if the predicted deviation between the novice and expert exceeds $\tau_m$, control is retained by the expert; otherwise, the novice is allowed to act. After collecting a batch of trajectories – including both policy-executed and expert-controlled segments – SafeDAgger performs the following training steps:

- The **policy** is updated via supervised learning on all states labeled by the expert, including those collected with noisy expert actions.
- The **doubt model** is trained as a binary classifier to predict whether the expert would be needed in a given state, using the collected trajectories and corresponding labels (policy vs. expert control).

The approach reduces the risk of unsafe behavior while still enabling data collection from the novice. By enforcing this margin-based constraint, SafeDAgger achieves a better trade-off between safety and exploration compared to the original DAgger formulation.

**LazyDAgger.**   LazyDAgger Hoque et al. (2021) is an extension of SafeDAgger that aims to reduce the frequency of context switches between the expert and the novice during data collection. Frequent switching can lead to unstable behavior, reduced training signal quality, and increased dependence on the expert. LazyDAgger introduces two key modifications to address these issues:

- **Hysteresis via dual thresholds:** Unlike SafeDAgger, which relies on a single threshold for the uncertainty predicted by a *doubt model*, LazyDAgger introduces two separate thresholds: a high threshold $\tau_h$ and a low threshold $\tau_l$, with $\tau_h > \tau_l$. These thresholds implement a *hysteresis* mechanism for control switching.
  Control is handed over to the expert when the predicted uncertainty exceeds $\tau_h$, and it is only returned to the policy once the uncertainty falls below $\tau_l$. This prevents rapid switching near the threshold boundary and results in more stable control delegation.

- **Noise injection in expert actions:** To improve the robustness and generalization of the learned policy, LazyDAgger injects noise into the expert's actions during data collection. This encourages the collection of more diverse trajectories and exposes the policy to a wider range of states, helping it to learn more effectively beyond deterministic expert behavior.

These modifications enable LazyDAgger to reduce the number of expert interventions required while improving data efficiency and policy robustness.

## 2.2 STOCHASTIC REACH-TUBE VERIFICATION OF DYNAMICAL SYSTEMS

Lagrangian reachability is a powerful technique for analyzing the evolution of dynamical systems under uncertainty. It constructs over-approximations of all states a system can reach over a finite time horizon by propagating sets of initial conditions through the dynamics. In the stochastic setting, these methods quantify uncertainty by incorporating probabilistic models of noise and disturbances, making them well-suited for verification and safety analysis from robotics to autonomous systems.

Stochastic Lagrangian Reachability (SLR) Gruenbacher et al. (2020) computes tight probabilistic reach-tubes for nonlinear systems by solving global optimization problems over disturbance realizations. These tubes serve as formal guarantees: they bound the probability that the system will remain within or exit a target set over time. In contrast to grid-based approaches like Hamilton-Jacobi reachability, which suffer from the curse of dimensionality, Lagrangian methods are more scalable and can be applied to higher-dimensional systems. Furthermore, they support both over- and under-approximations of reach-avoid sets, which are crucial in applications ranging from the traditional collision avoidance and spacecraft docking Gleason et al. (2017), to the emerging field of neural-network-control verification.

**GoTube.**   *GoTube* Gruenbacher et al. (2022) is a recent approach that brings Lagrangian reachability to deep learning pipelines. It constructs *stochastic reach-tubes* for black-box models, including neural network policies, by propagating probabilistic input distributions through nonlinear dynamics and learned components. GoTube achieves this by combining Lipschitz bounds, Gaussian approximations, and randomized smoothing techniques to efficiently estimate forward reachable sets under uncertainty.

Unlike traditional verification tools that require full system linearization or access to gradients, GoTube is designed to work with non-differentiable or opaque policies, making it highly practical for modern learning-based systems. It outputs tubes that contain the probabilistic future behavior of the system with a prescribed confidence level, enabling anticipatory safety decisions. As such, GoTube enables robust planning and control under uncertainty, and it is as a consequence particularly attractive for use in safety-critical learning pipelines. In this work, we leverage GoTube to construct a tube bounding states visited by a learned expert policy. We leverage this knowledge of typical states visited by the expert, by incorporating it into an interactive imitation learning loop to identify states visited by the novice requiring expert intervention.

Formally, stochastic reach-tube verification constructs a probabilistic enclosure around a system's possible trajectories over time, providing statistical guarantees on system behavior. Given an initial set of states $B_0$, the method generates a bounding tube, a sequence of bounding sets that stochastically encapsulates all potential states with a predefined confidence level $1 - \gamma$ with $1 > \gamma > 0$. The core of this approach relies on computing the maximum perturbation $\delta_t$ at each timestep:

$$\delta_t \geq \max_{x \in B_0} ||\chi(t, x) - \chi(t, x_0)|| = \max_{x \in B_0} d_t(x) \tag{2}$$

where $x_t$ is the system state at time $t$, starting from the initial point $x \in B_0$ with being the ball containing all initial states centered at $x_0$. The key challenge is ensuring the resulting bounding tube remains tight, avoiding over-approximation, while maintaining conservativeness within a desired confidence interval. To achieve this, stochastic reach-tube methods leverage Lipschitz continuity. Traditional interval-based techniques compute worst-case Lipschitz constants, leading to overly conservative bounds. Instead, stochastic Lipschitz caps are derived by bounding the local Lipschitz constants with a quantile-based stochastic lower bound $\Delta\lambda_{x,V}$. The radius of each stochastic cap is computed as in equation 3.

$$r_x = \frac{-\lambda_x + \sqrt{\lambda_x^2 + 4 \cdot \Delta\lambda_{x,V} \cdot (\mu \cdot \bar{m}_{t,V} - d_t(x))}}{2 \cdot \Delta\lambda_{x,V}} \tag{3}$$

where $V$ is the set of sampled traces, $\bar{m}_{t,V}$ is the maximum perturbation of those at time $t$, and $\mu$ is the tightness factor controlling the tube's balance between conservativeness and accuracy. The algorithm iteratively expands the set of sampled trajectories, refining the stochastic caps until the total surface coverage achieves the desired confidence level $\gamma$. Convergence of the method is guaranteed by Theorem 2 of Gruenbacher et al. (2022). Equation 4 states accordingly, that for every $\gamma$, there exists an $N = |V|$ such that the probability of any sample exceeding the true maximum perturbation $m_t^*$ is smaller than $\gamma$.

$$\forall\gamma \in (0, 1), \exists N \in \mathbb{N} \text{ s.t. } \Pr(\mu \cdot \bar{m}_{j,V} \leq m_j^*) \geq 1 - \gamma \tag{4}$$

This ensures the algorithm terminates with a valid bounding tube, even under stochastic perturbations. Unlike deterministic reachability methods, which suffer from wrapping effects and compounding over-approximation errors, stochastic reach-tube verification maintains stability over long time horizons by statistically bounding perturbations at each time step independently. The result is a scalable, probabilistically sound verification approach that supports high-dimensional, continuous-time systems. The approach particularly allows verifying systems governed by nonlinear ordinary differential equations (ODEs) and controlled by neural networks.

Safety during interaction with the environment when training the novice policy was the main concern that led to the development of many DAgger variants. The stochastic reach-tube verification approach outlined above can be used to classify the safety of a state and/or action. This means that we can replace the doubt model found in SafeDAgger with a Lagrangian reach-tube that was computed from expert trajectories beforehand. We call the resulting algorithm TubeDAgger. The advantage of this approach is that the criterion does not require any tuning of environment-dependent thresholds, given a reasonably tight reach-tube was computed beforehand.

## 3 TUBEDAGGER

We will now introduce TubeDAgger, a novel interactive imitation learning algorithm based on LazyDAgger, that employs stochastic reachtubes for the switching between autonomous and expert control. For this, we first compute a reachtube from a dataset of expert trajectories using an existing reachability tool. We then deploy a novice policy alongside an expert to learn from the experts' demonstrations. In order to achieve a robust novice policy, we allow the novice to take control from the expert when the current state is deemed safe. We consider a state to be safe if it is well inside the reachtube (a more formal definition of this requirement follows below).

---

**Algorithm 1** TubeDAgger

---

**Require:** Expert policy: $\pi^*(a|s)$, novice policy: $\hat{\pi}_\theta(a|s)$, intervention thresholds $\beta_+, \beta_-$, dataset of expert trajectories $\mathcal{D}$, reach-tube for expert: $(\mathcal{C}, \mathcal{R}, \mathcal{A})$, noise strength $\sigma^2$

    **while** not converged **do**

        $s_0 \leftarrow$ Initialize environment state

        Mode $\leftarrow$ Autonomous

        **for** $t = 0$ to $T$ **do**

            **if** Mode = Supervisor or $||A_t(s_t - c_t)^\top||_2 > r_t\beta_+$ **then**

                $a_t^* = \pi^*(s_t)$

                $\mathcal{D} \leftarrow \mathcal{D} \cup \{(s_t, a_t^*)\}$

                $s_{t+1} \leftarrow$ Execute $\tilde{a}_t^* \sim \mathcal{N}(a_t^*, \sigma^2 I)$

                **if** $||A_t(s_t - c_t)^\top||_2 \, r_t^{-1} < \beta_-$ **then**

                    Mode $\leftarrow$ Autonomous

                **else**

                    Mode $\leftarrow$ Supervisor

                **end if**

            **else**

                $s_{t+1} \leftarrow$ Execute $\hat{\pi}_\theta(s_t)$

            **end if**

        **end for**

        $\theta \leftarrow \arg\min_\theta \mathbb{E}_{(s_t, \hat{\pi}(s_t)) \sim \mathcal{D}}\left[\mathcal{L}\left(\pi^*\left(s_t\right), \hat{\pi}\left(s_t, \theta\right)\right)\right]$

    **end while**

---

We compute the tube for bounding observations using the GoTube Gruenbacher et al. (2022) package available on GitHub. With GoTube, we can generate a reach-tube given the expert controller and system. The tube contains information about what sequence of states the expert is likely to visit. When a state is outside the tube, it can be considered unsafe. We use this as the criterion for expert intervention in LazyDAgger. We assume that the closer a state is to the center of the tube, the more confident is the expert about successfully controlling the system. Therefore, we introduce a safety margin and hand back control to the expert when the distance of the current state to the tube center is larger than $\beta_+$ times the tube radius. Note that the tube criterion fully replaces the need for a doubt prediction model like in LazyDAgger.

The tube is given as a sequence of $(c, r, A, \tau)_t$ where $\tau$ is the time elapses since the episode start and $c$ and $A$ are the center of the tube and $A$ the metric defining the bounding ellipsoid at that time, respectively. Together, they can be used to check states for inclusion in the ellipsoid defined by using the affine transformation described by the metric and center. The current state $s_t$ is mapped to the unit circle using the inverse transformation of the ellipsoid $A_t(s_t - c_t)$. If the transformed state lies within the unit ball, the state before transformation is included in the ellipsoid tube. States that lie outside of the tube are considered unsafe and require training. In this case, the expert action is computed and executed. Samples with states that violate the safety condition are appended to the dataset. Algorithm 1 outlines the TubeDAgger approach. After $T$ steps in the environment, the novice policy is trained using the aggregated dataset. The loss function used is the mean squared error between expert action and novice prediction.

### 3.1 LIMITATIONS

One noteworthy limitation of TubeDAgger is that it requires knowledge of the temporal alignment of the system within a trajectory. Specifically, we need to know which $c_t$, $A_t$ and $r_t$ correspond to the current state $s_t$. Future work will explore dynamic time alignment methods, such as particle filtering, and assess the method's scalability to more complex robotic platforms and real-world domains.

It is also important to note, that the benefits discussed in this paper depend a lot on the quality of the reach-tube that is constructed beforehand. If the tube is too narrow, the algorithm will deteriorate to behavioral cloning; if it is too conservative, the training process will converge prematurely resulting in a sub-optimal novice policy. Computing a reach-tube for high-dimensional systems can be computationally expensive depending on the algorithm used. However, the tube needs to be computed only once - before the start of the imitation learning.

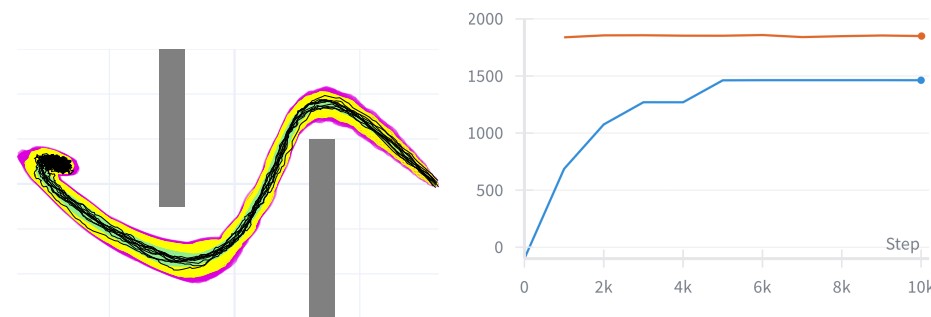

Figure 2: Reachtube for a 2D navigation toy example. The agent start on the right has to reach the goal position on the left while avoiding the gray walls. Left: The tube is depicted in purple; yellow and green respectively show the 0.7 and 0.2 boundaries used by TubeDAgger. Right: Reward curves for the imitator evaluation reward (blue) and the reward achieved by the combined imitator-expert agent (orange).

## 4 EXPERIMENTS

### 4.1 2D NAVIGATION TOY EXAMPLE

In order to demonstrate how TubeDAgger works in practice, we devised a simple 2D navigation toy example depicted in figure 2. Here, the task is to navigate from the starting position on the right to the goal position on the left – without crashing into the walls. We trained an expert policy RTRRL Lemmel & Grosu (2025) and then computed the reachtube from collected evaluation rollouts. Subsequently, we trained a novice policy using TubeDAgger. The reward curves that resulted from the training process are decpicted in figure 2 on the right. The straight orange line on the top shows how safety is ensured at all times during data collection using the combined imitator-expert agent.

### 4.2 MUJOCO PHYSICS SIMULATION

We further test our TubeDAgger approach empirically, on a set of continuous control tasks commonly known as Mujoco environments. MuJoCo (Multi-Joint dynamics with Contact) is a fast, accurate physics engine for simulating multi-body systems, widely used in robotics, biomechanics, and reinforcement learning (RL). It supports customizable environments with realistic physical interactions and is a go-to tool for RL research. Brax is a lightweight reimplementation of MuJoCo in JAX, designed for fast, GPU/TPU-accelerated simulations. It offers similar environments and is optimized for scalable RL training and differentiation. Brax also provides reference implemenations of state-of-the-art reinforcement learning algorithm, and tuned hyperparameters for every environment facilitating training of expert policies. We use the environments included in the brax package for the evaluation of TubeDAgger in simulation. The environments are: `inverted_pendulum`, `inverted_double_pendulum`, `ant`, and `halfcheetah`. We trained an expert policy for each environment using the provided PPO implementation.

We implemented a wrapper for expert policies to be used by GoTube, to compute a reach-tube encapsulating typical observations. Additionally, we constructed a second kind of tube with additional dimensions for the action added. The hyperparameter setting we used for all tasks were: $\gamma = 0.2$, $\mu = 1.0$, ellipsoids=True, radius= 0.01 and batch_size= 512. Figure 5 shows plots of the first two system dimensions for some of the generated tubes.

Having computed the necessary tubes, we ran imitation learning experiments with our TubeDAgger algorithm, using both sets of tubes described above. We also recorded experiments with LazyDAgger as a baseline. For all, we ran 5 runs using different thresholds in each environment. The results are shown in Figures 7, 8 and 9.

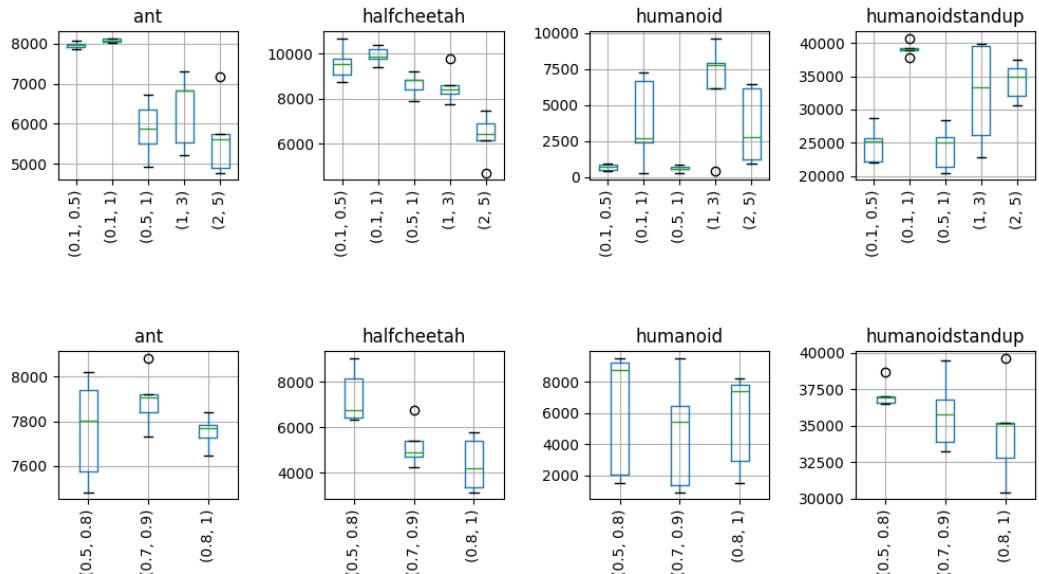

Figure 3: Boxplots showing the validation rewards for 5 runs each of LazyDAgger and TubeDAgger with different lower and upper thresholds for the action distance. Top row shows results for LazyDAgger and bottom row for TubeDAgger. When comparing to the LazyDAgger results above, we can see that TubeDAgger is more robust to the choice of threshold.

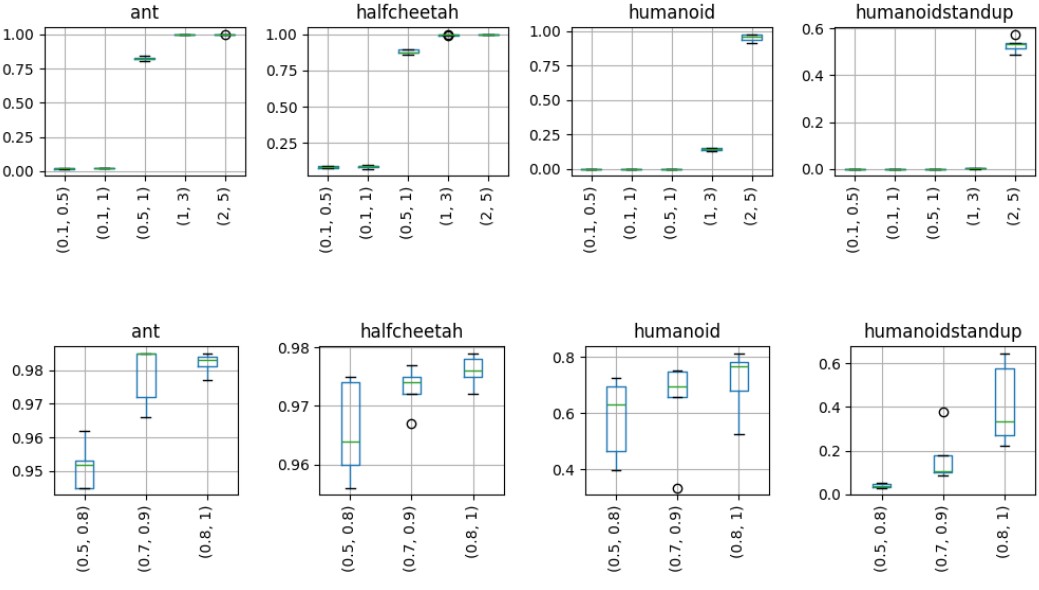

Figure 4: Boxplots showing the percentage of novice actions at the end of training for 5 runs each with different lower and upper thresholds. The top row shows LazyDAgger and the bottom row TubeDAgger results. Again, we can see improved robustness to hyperparameter choice when compared to LazyDAgger.

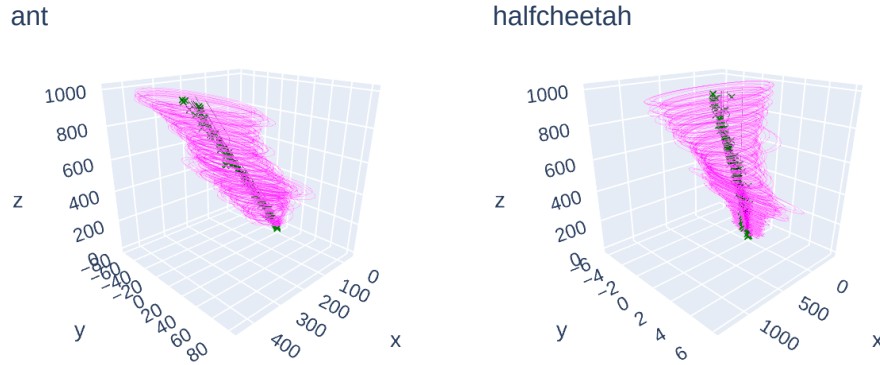

Figure 5: Reach-tubes generated by GoTube. The plots show the first two dimensions of the respective system as x- and y-axes with the z-axis denoting time.

Table 1 shows the validation rewards achieved by the trained novice policies in comparison to the expert policies. The reported numbers are for the best thresholds found during the hyperparameter sweep. Tables 3 and 4 in the Appendix show the complete results.

Finally, we depict the highest percentage of policy actions in Figure 10 as a boxplot, and as line plots for two selected environments in Figure 11. We further show the number of context switches for LazyDAgger vs TubeDAgger in table 6. These figures are from models trained with bounds $(0.1, 1)$ and $(0.5, 0.8)$ for LazyDAgger und TubeDAgger respectively. Especially for the inverted pendulum environment, the number of context switches was reduced drastically.

| Environment | LazyDAgger | EnsembleDAgger | TubeDAgger |
|---|---|---|---|
| ant | 8013.46±103.36 | 8065.50±38.69 | 7739.51±139.28 |
| halfcheetah | 10197.03±403.71 | 10789.82±111.25 | 9134.71±732.20 |
| inverted double pendulum | 8651.46±615.54 | 8665.62±503.88 | 9359.60±377.46 |
| inverted pendulum | 1000.00±0.00 | 1000.00±0.00 | 1000.00±0.00 |

Table 1: Comparison of evaluation rewards across algorithms for different environments. Given are the median reward and standard deviation for five runs each of the best respective hyperparameter configuration. It has to be noted that EnsembleDAgger used $5\times$ the number of parameters.

## 5 RELATED WORK

### 5.1 INTERACTIVE IMITATION LEARNING

DAgger provides formal guarantees on policy performance but requires significant expert supervision. Various extensions of DAgger have been proposed, including SafeDAgger Zhang & Cho (2017), which uses a safety predictor to reduce expert queries, and EnsembleDAgger Menda et al.

| Environment | LazyDAgger | EnsembleDAgger | TubeDAgger |
|---|---|---|---|
| ant | 242.00±200.98 | 219.80±54.88 | 186.50±34.85 |
| halfcheetah | 536.20±357.23 | 109.20±131.45 | 84.00±4.24 |
| inverted double pendulum | 2327.20±1214.42 | 0.00±0.00 | 0.00±0.00 |
| inverted pendulum | 1426.40±221.35 | 283.60±58.03 | 31.60±14.52 |

Table 2: Supervisor burden as in Hoque et al. (2021) with cost for switching and per step assumed as 1 and 0.1 respectively.

(2019), which leverages uncertainty estimates from model ensembles to guide expert intervention. Interactive learning methods reduce the burden on experts by querying them selectively. LazyDAgger Hoque et al. (2021) introduced a doubt-based intervention mechanism that requests expert feedback only when the learner's confidence falls below a threshold, requiring a separate doubt classification model. Similarly, DART Laskey et al. (2017) adds noise during expert demonstrations to better cover states where the learner might make mistakes.

Active learning approaches further minimize expert interactions by querying only the most informative states. CEIL Cheng et al. (2018) employs importance sampling to focus expert feedback on difficult states. HG-DAgger Kelly et al. (2019) uses human gaze data to identify critical states requiring expert attention. Our work differs from these approaches by leveraging reachability analysis rather than classification or uncertainty measures to determine when expert intervention is necessary.

Safety guarantees are crucial in robotics and autonomous systems. Risk-Sensitive Generative Adverserial Imitation Learning Lacotte et al. (2019) optimizes policies considering the risk of constraint violations. SafetyNet Vitelli et al. (2022) uses neural networks to predict and avoid unsafe actions. Recently, a shielding approach based on control-barrier functions and inverse reinforcement learning was introduced, that aims at resampling actions in order to find a safe continuation Yang et al. (2024). Another recent approach called RACER Dai et al. (2024) incorporates language models to generate recovery plans in robotics applications. Our work combines elements from these safety-critical approaches with interactive imitation learning.

## 5.2 REACHABILITY ANALYSIS AND STOCHASTIC REACHTUBES

Reachability analysis computes the set of states a system can reach over time. Deterministic reachability has been extensively studied for verification of hybrid systems Althoff (2015), but is often too conservative for practical use in uncertain environments. Stochastic reachability extends this concept to systems with probabilistic dynamics Summers & Lygeros (2010). Stochastic reachtubes represent the probability distribution of future states given the current state and action. Recent work has developed computationally efficient methods for approximating these tubes, including approaches based on Hamilton-Jacobi reachability analysis Bansal et al. (2017), Gaussian processes Jackson et al. (2020), and neural network surrogates Hashemi et al. (2023). Our TubeDAgger algorithm builds upon these advances, using stochastic reachtubes to make informed decisions about when expert intervention is necessary.

## 5.3 LEARNING WITH MINIMAL EXPERT SUPERVISION

Several approaches aim to minimize expert supervision in learning complex behaviors. Methods like TAMER Knox & Stone (2009) and COACH MacGlashan et al. (2017) incorporate human feedback into the learning process. One-shot imitation learning Duan et al. (2017) attempts to generalize from a single demonstration. Semi-supervised approaches like PLATO Kahn et al. (2017) combine small amounts of expert data with larger unlabeled datasets.

Most closely related to our work is LazyDAgger Hoque et al. (2021), which selectively requests expert intervention. However, LazyDAgger relies on a doubt classification model that must be trained alongside the policy, potentially introducing additional complexity and failure modes. TubeDAgger addresses this limitation by replacing the doubt classifier with stochastic reachtubes, providing a more principled approach to determining when expert intervention is necessary while maintaining theoretical guarantees on policy performance.

## 6 CONCLUSIONS

We introduced TubeDAgger, an interactive imitation learning algorithm that leverages stochastic reach-tube verification to determine safe regions for novice control. Unlike previous methods that rely on manually tuned doubt models or distance thresholds, TubeDAgger uses precomputed reachtubes to define safety regions with probabilistic guarantees, making it more principled and generalizable. Our results on simulated Mujoco environments demonstrate that TubeDAgger achieves comparable or better performance than LazyDAgger while requiring fewer expert interventions.

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

## A    TECHNICAL APPENDICES AND SUPPLEMENTARY MATERIAL

### A.1    HARDWARE

All experiments were run on a NVIDIA RTX A6000 GPU. Each imitation learning run took up to 15 minutes and a total of around 1000 runs were made for this work. Time needed for computing the reach-tubes ranged from ten minutes to three hours, depending on the number of dimensions of the system.

### A.2    PARAMETER SWEEP RESULTS

## B    A DETAILED DESCRIPTION OF GOTUBE

Table 3: Validation reward for LazyDAgger with different thresholds $(\beta_-, \beta_+)$.

| env name
distance expert | ant | halfcheetah | humanoid |
|---|---|---|---|
| (0.1, 1) | 8017.84±118.03 | 9676.12±513.90 | 2275.57±2603.89 |
| (0.1, 0.5) | 7962.76±79.77 | 9560.95±741.55 | 683.50±235.85 |
| (0.5, 1) | 5884.67±706.56 | 8645.16±515.87 | 588.51±202.57 |
| (1, 3) | 5473.15±1174.49 | 7975.40±1018.68 | 3606.54±3754.95 |
| (2, 5) | 5241.45±1344.84 | 5536.32±1373.67 | 2807.24±2064.10 |

| env name

distance expert | humanoidstandup | inverted double pendulum | inverted pendulum |
|---|---|---|---|
| (0.1, 1) | 30707.32±9085.59 | 8863.28±488.98 | 1000.00±0.00 |
| (0.1, 0.5) | 24749.00±2776.67 | 8953.03±372.75 | 1000.00±0.00 |
| (0.5, 1) | 24186.32±3263.30 | 2692.36±732.63 | 1000.00±0.00 |
| (1, 3) | 27576.50±7418.80 | 1927.51±2130.59 | 709.83±467.23 |
| (2, 5) | 31118.04±4171.12 | 267.88±113.42 | 23.64±9.68 |

Table 4: Validation reward for TubeDAgger with different thresholds $(\beta_-, \beta_+)$ where the tube was only computed for the observations.

| env name
tube | ant | halfcheetah | humanoid |
|---|---|---|---|
| (0.9, 1) | 4319.88±4066.52 | -340.44±8029.05 | 1347.41±3892.09 |
| (0.5, 0.8) | 7763.23±231.48 | 7338.22±1207.44 | 6225.94±4050.18 |
| (0.8, 1) | 7754.41±72.98 | 4365.11±1190.94 | 5564.08±3123.07 |
| (0.7, 0.9) | 7896.50±128.21 | 5191.67±970.89 | 4750.66±3614.40 |

| env name
tube | humanoidstandup | inverted double pendulum | inverted pendulum |
|---|---|---|---|
| (0.9, 1) | 18475.72±18429.99 | 5576.68±4593.79 | 510.85±515.63 |
| (0.5, 0.8) | 37125.20±902.68 | 9081.94±383.73 | 1000.00±0.00 |
| (0.8, 1) | 34638.83±3405.72 | 9084.03±378.35 | 1000.00±0.00 |
| (0.7, 0.9) | 35822.89±2461.51 | 9082.89±382.14 | 1000.00±0.00 |

The sequence of $(c_t, r_t, A_t)$ is computed using the Go-Tube algorithm. The algorithm starts from a ball of initial states defined by center point $c_0$ and radius $r_0$ (Note that since it starts from a ball $A_0$ is the unit matrix). It proceeds by iteratively sampling a batch of initial states from the surface of the ball, and requesting expert rollouts. For each step, an ellipsoid - defined by matrix $A_t$ together with a radius $r_t$ and center $c_t$ - is chosen in a way, that ensures that all the states $\mathcal{S}_t$ reachable by the expert are contained with probability $1 - \gamma$. Specifically, $A_t$ is computed from the set of states by rotating using

| Parameter Name | Value |
|---|---|
| $\gamma$ | 0.2 |
| $\mu$ | 1.0 |
| ellipsoids | True |
| radius | 0.1 |
| batch_size | 512 |

Table 8: GoTube hyperparameter settings used for generating reachtubes.

PCA components and then scaling the data points; $A_t$ represents the combined transformation that makes all the states lie within the unit circle. We refer to the original GoTube paper for details on how the target probability is guaranteed. It can happen that the probability can not be met with the initial set of trajectories, in which case another batch of expert trajectories is requested.

Table 8 shows the hyperparamters that were used for computing the reachtubes used in this work.

Table 5: Validation reward for TubeDAgger with different thresholds $(\beta_-, \beta_+)$ where the action was included in the tube.

| env name
tube | ant | halfcheetah | humanoid |
|---|---|---|---|
| (0.9, 1) | -764.86±329.67 | -855.54±3362.10 | -1384.55±1546.98 |
| (0.5, 0.8) | 7916.78±182.73 | 9217.55±746.29 | 3234.53±1942.81 |
| (0.8, 1) | 8041.32±45.40 | 8894.91±856.02 | 5104.33±4109.95 |
| (0.7, 0.9) | 7936.43±113.75 | 9327.90±381.18 | 2751.54±3738.62 |
| env name
tube | humanoidstandup | inverted double pendulum | inverted pendulum |
| (0.9, 1) | 11182.78±9494.46 | 281.59±151.54 | 21.70±7.44 |
| (0.5, 0.8) | 37447.19±2266.80 | 9081.94±383.73 | 1000.00±0.00 |
| (0.8, 1) | 36009.31±1381.47 | 9081.94±383.73 | 1000.00±0.00 |
| (0.7, 0.9) | 33010.53±7481.98 | 9081.94±383.73 | 1000.00±0.00 |

| env name | LazyDAgger | EnsembleDAgger | TubeDAgger |
|---|---|---|---|
| ant | 171.50±186.61 | 235.00±27.63 | 10.00±6.16 |
| halfcheetah | 316.00±151.09 | 66.00±34.01 | 30.50±7.20 |
| humanoid | 602.00±19.07 | NaN | 49.00±4.87 |
| humanoidstandup | 643.00±125.40 | NaN | 61.00±2.55 |
| inverted double pendulum | 118.00±186.12 | 188.00±1.58 | 0.00±0.00 |
| inverted pendulum | 94.00±202.12 | 155.00±22.92 | 0.00±0.00 |
| pusher | NaN | 323.00±7.77 | NaN |
| reacher | NaN | nan±nan | NaN |

Table 6: The number of context switches until solved.

| env name | LazyDAgger | EnsembleDAgger | TubeDAgger |
|---|---|---|---|
| ant | 0.00±7.08 | 1.00±0.89 | 2.00±1.44 |
| halfcheetah | 1.00±5.58 | 0.00±0.89 | 3.50±4.47 |
| humanoid | 1.00±15.69 | NaN | 1.00±0.84 |
| humanoidstandup | 0.00±2.38 | NaN | 0.00±0.44 |
| inverted double pendulum | 1.00±3.79 | 0.00±0.00 | 0.00±0.00 |
| inverted pendulum | 2.00±4.12 | 1.00±1.14 | 0.00±0.00 |
| pusher | NaN | 0.00±0.00 | NaN |
| reacher | NaN | nan±nan | NaN |

Table 7: The total number of expert actions until solved.

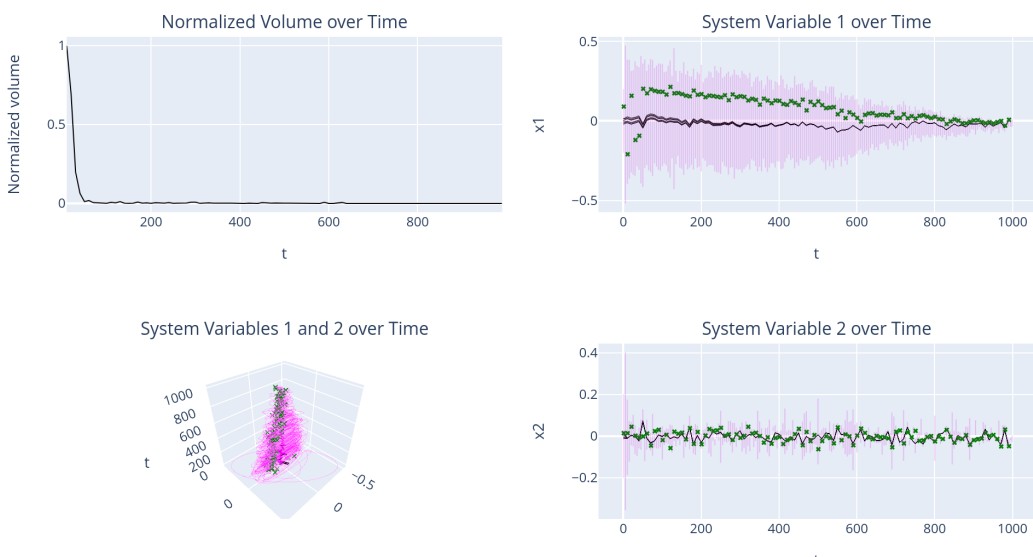

Figure 6: A plot as created by GoTube that shows the reach-tube's normalized volume over time and the first two system variables. Green crosses show the sample with the largest distance (over all dimensions) to the center at this step in time.

## B.1 SAFETY GUARANTEES

After training an imitator, we can also compute a reachtube for the resulting policy. We then can check for containment in the initial tube to obtain a safety guarantee. If the time-indexed sequence of sets - that cover all possible imitator traces with probability $p$ - are all contained within the corresponding sets of the initial tube, then the tube is as safe as the expert policy with probability $p$.

Formally, if $\mathcal{T}^\pi = \{T_0^\pi, T_1^\pi, \ldots, T_K^\pi\}$ is the time-indexed sequence of sets (the reachtube) for the imitator policy $\pi$, such that

$$\Pr\left(x_j^\pi \in T_j^\pi \ \forall j = 0, \ldots, K\right) \geq p,$$

and if the expert (reference) tube is $\mathcal{T}^E = \{T_0^E, T_1^E, \ldots, T_K^E\}$, then the safety guarantee can be stated as:

$$\text{If } \forall j \in \{0, \ldots, K\} : T_j^\pi \subseteq T_j^E, \text{ then } \mathcal{T}^\pi \text{ is as safe as } \mathcal{T}^E \text{ with probability } p,$$

i.e.,

$$\left[\forall j : T_j^\pi \subseteq T_j^E \text{ and } \Pr\left(x_j^\pi \in T_j^\pi \ \forall j\right) \geq p\right] \implies \text{ imitator as safe as expert with probability } p.$$

## C ADDITIONAL RESULTS

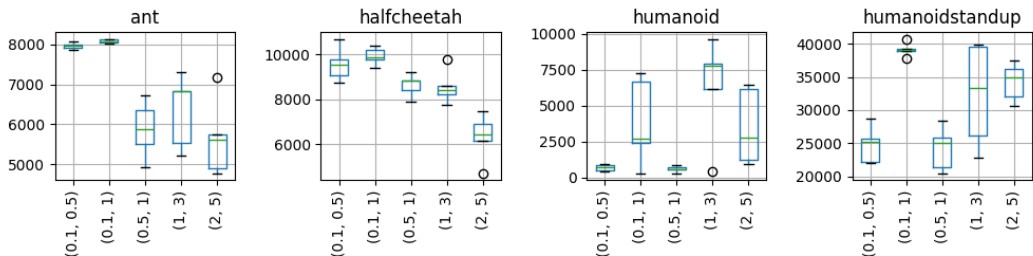

Figure 7: Boxplots showing the validation rewards for 5 runs each of LazyDAgger with different lower and upper thresholds for the action distance.

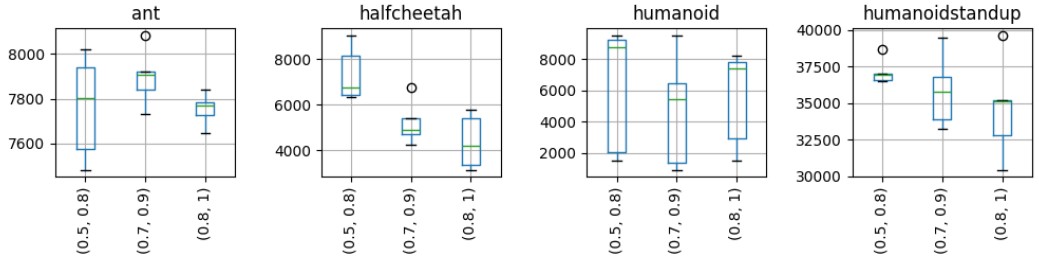

Figure 8: Boxplots showing the validation rewards for 5 runs each of TubeDAgger with different lower and upper thresholds for the distance from the tube center. When comparing to the LazyDAgger results above, we can see that TubeDAgger is more robust to the choice of threshold.

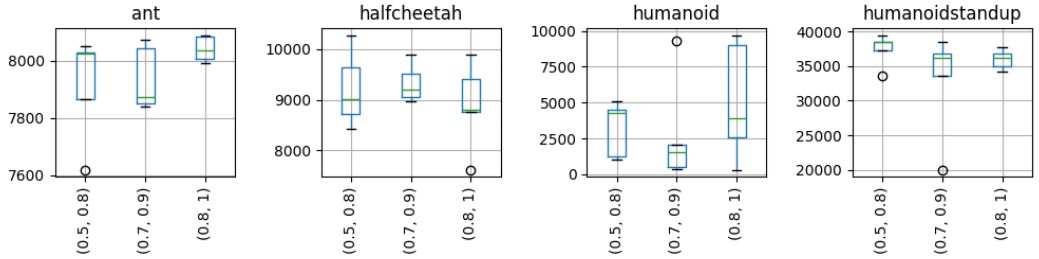

Figure 9: Boxplots showing the validation rewards for 5 runs each of TubeDAgger with different lower and upper thresholds for the distance from the tube center. When comparing to the LazyDAgger results above, we can see that TubeDAgger is more robust to the choice of threshold.

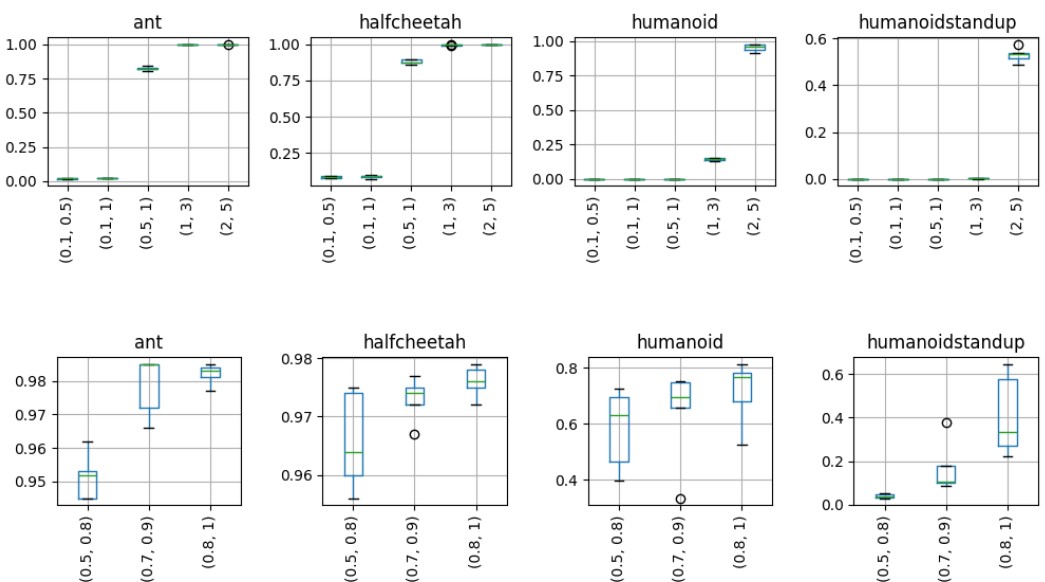

Figure 10: Boxplots showing the percentage of novice actions at the end of training for 5 runs each with different lower and upper thresholds. The top row shows LazyDAgger and the bottom row TubeDAgger results. Again, we can see improved robustness to hyperparameter choice when compared to LazyDAgger.

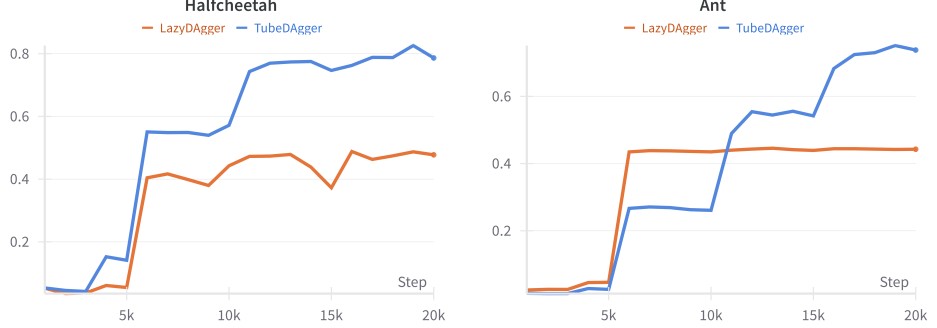

Figure 11: The mean percentage of novice actions aggregated over all our runs.

