# OpenReview forum: "TubeDAgger: Reducing the Number of Expert Interventions with Stochastic Reach-Tubes"
_ICLR.cc/2026/Conference — ICLR 2026 Conference Withdrawn Submission_

### Official Review · Reviewer_QnHu · 2025-10-24

**Soundness:** 2
**Presentation:** 3
**Contribution:** 2
**Rating:** 2
**Confidence:** 4

**Summary:**

The paper proposes a new interactive imitation learning algorithm TubeDAgger. TubeDAgger constructs a reach-tube from expert demonstrations using Stochastic Lagrangian Reachability, and it uses the boundary of this reach-tube as the condition for querying the expert. Experimentally, the authors claim that TubeDAgger outperforms both LazyDAgger and EnsembleDAgger on MuJoCo tasks.

**Strengths:**

1. The paper is the first one that applies Stochastic Lagrangian Reachability (SLR) to interactive imitation learning.
2. TubeDAgger is highly intuitive and simple to implement, and the reach tube in Fig. 1 provides strong interpretability for this algorithm.

**Weaknesses:**

1. The paper **only includes LazyDAgger and EnsembleDAgger as the baselines** in Section 4.
The current state-of-the-art DAgger-based IIL baseline is **ThriftyDAgger** [1]. ThriftyDAgger combines uncertainty estimates from model ensembles and risk estimates from a learned Q-function as the criterion of expert queries. Also, **HG-DAgger** [2] improves upon EnsembleDAgger by asking for human help when the average uncertainty of the most recent 25 percent of expert interventions is large. Notably, ThriftyDAgger has good performance on complex **real-world** tasks such as peg insertion and cable routing, which is stronger than the 2D navigation and Mujoco environments in this paper.
In addition, there are many interactive imitation learning methods that are not based on DAgger, including Intervention Weighted Regression [3], PVP [4], and Shared Autonomy via Deep Reinforcement Learning [5]. These approaches have strong results on tasks involving autonomous driving, drone control, and robotics.

2. The paper does **not** explain how to **construct reach-tubes efficiently for environments with domain randomization**.
For tasks with multiple scenarios, such as varying goal or start positions or stochastic env noises, the reach-tube would need to be reconstructed for each case. So the data required by TubeDAgger will scale linearly with the number of env scenarios, harming TubeDAgger's generalizability to autonomous driving or real-world robotics.
In contrast, ThriftyDAgger has solved the peg insertion task without requiring additional expert data for every initial peg position. I recommend the author to report TubeDAgger's performance on more Mujoco tasks such as Reacher and Pusher, which involve **randomized initial states and targets**.

3. There's **no** experiments involving **real human or real-world tasks**. All experiments are conducted in simulation with PPO experts approximating humans. However, real human demonstrations and real-world deployment have significantly greater uncertainty.

References:
[1] ThriftyDAgger: Budget-Aware Novelty and Risk Gating for Interactive Imitation Learning, Ryan Hoque et al., 2021
[2] HG-DAgger: Interactive Imitation Learning with Human Experts, Michael Kelly et al., 2019
[3] Human-in-the-Loop Imitation Learning using Remote Teleoperation, Ajay Mandlekar et al. 2020
[4] Learning from Active Human Involvement through Proxy Value Propagation, Zhenghao Peng et al. 2023
[5] Shared Autonomy via Deep Reinforcement Learning, Siddharth Redd et al. 2018

**Questions:**

1. In Table 1, **how many env interactions and expert samples** are used? The paper does not mention whether TubeDAgger and the baselines used the same number of environment interactions or expert queries. It seems that TubeDAgger does not outperform baselines in Table 1 (ant, TubeDAgger 7739.51 vs LazyDAgger 8013.46, etc).
2. Why are Figures 3 and 4 comparing different hyperparameter choices instead of sample budgets? It's better to compare the evaluation reward under the same expert sample budget. Alternatively, could the authors plot eval performance and expert query count against the number of environment interactions to show TubeDAgger's training efficiency?

---

### Official Review · Reviewer_ZUah · 2025-10-30

**Soundness:** 1
**Presentation:** 2
**Contribution:** 2
**Rating:** 2
**Confidence:** 4

**Summary:**

The paper proposes a method to improve upon DAgger in the setting of Interactive Imitation Learning. The method combines stochastic reachability analysis with DAgger to reduce expert interventions without the need for a separate classifier for identifying safety regions (as seen in prior works).

Overall, the motivation of the method is clear, and the effort to reduce expert interventions in online imitation learning appears to be an exciting area of research. However, the contribution of the paper seems to be a simple combination of methods and requires further justification to be deemed novel. Experiments are conducted in simulation environments, where the method is compared to other DAgger-based approaches. The results are not very convincing and require further explanation.

**Strengths:**

* The motivation of the method is clear, and the effort to reduce expert interventions in online imitation learning appears to be an exciting area of research
* The idea of combining traditional reachability analysis with a learning-based approach seems to be interesting
* The method section is mostly easy to follow, and the related work section provides good insights into how this work sits among prior works

**Weaknesses:**

* First, the novelty of the paper needs to be improved. As currently presented, it seems the method leverages a reachability analysis approach (GoTube) as a black-box method and simply combines it with LazyDAgger. The modifications beyond existing approaches are minimal.
* The limitation section poses relevant questions that should be answered by the authors in the paper: for example, how the authors decide whether a constructed reach-tube has good quality in their experiments.
* The experiment results are not convincing. First, from Table 1, TubeDAgger underperforms in 2/4 environments and outperforms in 1/4 environment. No further explanation is provided by the authors regarding the results.
* The example in Section 4.1 does not demonstrate the need for stochastic reach-tubes. There is a significant performance gap between the expert and novice policies, and it is unclear why the stochastic tube is helping with learning.
* The experiment section is poorly organized. Table 2, Figure 3, and Figure 4 are in the main paper but are never referenced; results from Table 1 are not explained.

**Questions:**

* What is the novelty of this work compared to exisiting approaches?
* What is the metric to measure the effectiveness of your method?
* What are the considerations on whether a reach-tube could be used for downstream imitation learning once it's constructed?
* What are the expert performances on these tasks?

---

### Official Review · Reviewer_cCUF · 2025-10-31

**Soundness:** 2
**Presentation:** 1
**Contribution:** 1
**Rating:** 2
**Confidence:** 4

**Summary:**

The paper introduces TubeDAgger, an interactive imitation learning approach based on the DAgger framework, where the expert can be queried for interventions to assist the agent. TubeDAgger replaces the doubt classifier of SafeDAgger with stochastic reach-tubes. These are computed with GoTube, a method able to extract reach-tubes from black box controllers, and are used to determine when expert intervention is needed. The criterion is based on the one devised by LazyDAgger.

**Strengths:**

* The topic of designing imitation learning methods that ensure safety constraints is timely and valuable.
* The diagram in Fig. 1 is very clear.

**Weaknesses:**

* Overall, the presentation requires a fair bit of polish to read smoothly.
* If my understanding is correct, there is a tube ellipsoid for each timestep in a trajectory, computed from the expert trajectory dataset. TubeDAgger considers the tube ellipsoid cut at timestep t when the agent (novice) is at timestep t. That is a clear limitation, since the novice is unlikely to be caught up with the expert at every timestep. The authors leave the alignment problem to future work. It is debatable whether that should not be treated already here because it considerably limits the applicability of the approach to pretty much any setting where the agent is not always aligned with the expert at every timestep.
* By the end, it is still unclear how getting a reach-tube from an algorithm like GoTube is better (results, but especially in cost) compared to learning a doubt classifier.
* More generally, it reads like a combination of methods and does not lay foundations in a compelling enough way to foster further research in that direction.

**Questions:**

* L160: “learned expert policy”? What does that mean? It that just means that the authors learned the expert policy in a preliminary step, the authors can here simply call it “expert policy” and refer the reader to the section where the protocol that was used to trained the expert is described.
* Eq. 2: what is X-cursive/Chi? Define d_t.
* L176: the authors should  explain the notations. At least add a reference to where it is defined and explained in the original work. Where does Eq. 3 comes from?
* The authors write that one advantage of TubeDAgger compared to SafeDAgger is that the criterion does not require the tuning of environment-dependent thresholds. But TubeDAgger still needs to precompute a good reach-tube. Is that any different in terms of compute cost, time, and/or engineering work? The authors should characterize how tuning the threshold is more tedious that crafting a good reach-tube. Is it not clear in the paper.
* L256: to clarify and ground the method in prior works and their nomenclature, could one say that this is behavioral cloning on the augmented dataset?
* L266: the authors write: “If the tube is too narrow, the algorithm will deteriorate to behavioral cloning”. What exactly to the authors refer to in that case? It is going to be behavioral cloning either way, only the dataset contents changes. My understanding is that if the tube is too narrow, the novice will go outside the safety region too often and the expert interventions will be very frequent, and we will end up with an abundant dataset. The authors also write: “if it is too conservative, the training process will converge prematurely resulting in a sub-optimal novice policy”. My understanding is that, in that case, the novice will never activate the intervention criterion and the novice will be trained only on the initial expert dataset, resulting in vanilla behavioral cloning in effect.
* The authors write: “Computing a reach-tube for high-dimensional systems can be computationally expensive depending on the algorithm use”. Since only GoTube is brought up here, it would be relevant to qualify how it fares against the other algorithms that compute tubes?
* L296-297: the authors write: “from collected evaluation rollouts”. How many rollouts? What is the length of those rollouts in terms of timesteps/transitions? What is the frequency of the simulation?
* The authors should characterize what is to be observed in table 5 in the text. From what I read, TubeDAgger gets worse than the baselines when the env DoFs increase.
* Table 2 is referred nowhere in the text. It is unclear what “superior burden” refers to and how to read the table. I am assuming it refers to “expert interventions”?
* The lower number of expert interventions displayed by TubeDAgger compared to the baselines is appealing, but we (readers) lack information about how many demonstrations are in use, which would ground the number of interventions, e.g. do the interventions represent 50% of the available data?

Style, typos, suggestions:
* Fig 2: it would maybe be clearer to present the colors as: orange = with TubeDAgger, blue = without TubeDAgger
* Figs 3 and 4 appear in the paper without being referenced in the text? Humanoid environments are not among the Brax list written in the text.
* Revise and unify emdashes formatting/appearance throughout the entire paper.
* It could help to be clearer about what “familiar” means in this context (L45).
* L106-107: phrasing and clarity is subpar; I suggest revising the line.
* Missing reproducibility statement section.
* Only 1 subsection in section 3
* Consistency across the paper: choose one of “reach-tube” or “reachtube”.
* L215: “control” -> “controls”?
* L239-242: could enhance style and reduce redundancies in the text.
* L248-249: “and c and A are the center of the tube and A the” -> error.
* L253: “require training” -> “require expert intervention”?
* Fig 2: it would be more appropriate to build the plots with a plotting library as is customary, as opposed to using a screenshot from a Weights and Biases dashboard like you do here. Also, missing legend on the plot.
* L298-299: “decpicted” -> typo.
* L318-319: prefer not using the “=“’s; besides, spacing around those is inconsistent.
* NaNs in tables in appendix. If those were purposely written there, the authors should justify why they obtain such results for the given methods, and why they did not manage to make the methods numerically stable.

---

### Official Review · Reviewer_Kvx7 · 2025-11-03

**Soundness:** 2
**Presentation:** 3
**Contribution:** 1
**Rating:** 2
**Confidence:** 4

**Summary:**

This paper addresses the practical challenge of minimizing the number of expert interactions in interactive imitation learning (IL) algorithms such as DAgger, where excessive expert query is costly and inefficient. Existing extensions like SafeDAgger and LazyDAgger attempt to reduce expert queries through a doubt model that predicts whether the novice’s action deviates too far from the expert’s. However, these methods suffer from several drawbacks: they introduce additional learning complexity, require environment-dependent threshold tuning, and remain sensitive to classification errors in the doubt model. To overcome these limitations, this paper propose TubeDAgger, a novel variant that replaces the learned doubt model with a stochastic Lagrangian reachability framework, implemented via GoTube. By precomputing probabilistic reach-tubes that encapsulate the expert’s reachable state distributions, TubeDAgger determines when to request expert intervention based on whether the novice’s state lies outside this tube.

**Strengths:**

1. The paper introduces a novel integration of stochastic Lagrangian reachability, through GoTube, into the interactive IL setting. This idea replaces the conventional doubt model–based safety estimation used in SafeDAgger and LazyDAgger, offering an efficient mechanism for expert intervention without additional training. The method is well-motivated and technically grounded in existing stochastic reachability theory.


2. The paper is clearly written, with the overall method pipeline and theoretical backgrounds (GoTube and stochastic reach-tube construction) well explained. Algorithmic components are presented in a self-contained manner, enabling reproducibility and conceptual transparency.


3. TubeDAgger removes the need for training a separate doubt model and thus simplifies the interactive IL pipeline substantially. Moreover, it exhibits robustness to hyperparameter choice, addressing one of the most critical limitations of prior DAgger variants. The proposed approach broadens the applicability of safe imitation learning to more complex and safety-critical domains, representing a step toward more scalable and reliable interactive IL frameworks.

**Weaknesses:**

**1. Strong dependency on previous approach (GoTube) with limited quantitative validation**

The core novelty of TubeDAgger heavily relies on GoTube, an existing stochastic reachability tool. While this integration is technically sound, the paper does not quantitatively analyze how well the computed stochastic reach-tubes replicate or improve upon the function of the doubt model used in SafeDAgger and LazyDAgger. For example, there is no systematic comparison of safety classification accuracy, intervention timing precision, or robustness under model mismatch. A dedicated ablation study isolating GoTube’s contribution would substantially strengthen the empirical claims.

**2. Lack of theoretical guarantees beyond probabilistic safety**

Similar to prior DAgger variants, TubeDAgger still remains a heuristic rather than a principled method. Although the reach-tube formulation provides a probabilistic safety bound, it does not yield any suboptimality or regret guarantee regarding the learned policy’s performance, which is originally guaranteed in DAgger. This gap limits the theoretical contribution and makes it difficult to assess whether the method achieves comparable convergence properties to the original DAgger framework.

**3. Incremental empirical improvement and limited experimental scope**

As shown in Table 1, the performance improvement over SafeDAgger and LazyDAgger appears incremental, mainly reflected in the reduced number of expert interventions, while the overall reward gains remain modest. Moreover, the experiments are too limited to a small set of continuous control benchmarks (e.g., MuJoCo), leaving open questions regarding the method’s scalability to high-dimensional or safety-critical domains such as robotics or autonomous driving. In addition, only two baselines are considered, which makes it difficult to assess the relative efficiency of the proposed approach. The absence of publicly available implementation or experimental details raises concerns about reproducibility.

**Questions:**

Q1. Could the authors report binary classification performance (e.g. precision/recall or ROC curves, …) comparing the safety classification of the reach-tube (safe vs. unsafe) against an oracle that uses expert disagreement as ground truth (or doubt model)?

Q2.  The paper’s main claim is the reduction in expert interventions, but this relationship is not visualized clearly. Would the authors consider including a scatter plot (x-axis: number of expert interactions, y-axis: normalized performance) to better illustrate efficiency–performance trade-offs?

---

### Note · Authors · 2025-11-30

I have read and agree with the venue's withdrawal policy on behalf of myself and my co-authors.